# Single-molecule force spectroscopy of protein-membrane interactions

Lu Ma[1,2,3†], Yiying Cai[1,4,5,6†], Yanghui Li[1,7], Junyi Jiao[1,8], Zhenyong Wu[9,10], Ben O'Shaughnessy[11], Pietro De Camilli[1,4,5,6,12]*, Erdem Karatekin[9,10,13,14]*, Yongli Zhang[1]*

[1]Department of Cell Biology, Yale University School of Medicine, New Haven, United States; [2]CAS Key Laboratory of Soft Matter Physics, Institute of Physics, Chinese Academy of Sciences, Beijing, China; [3]Beijing National Laboratory for Condensed Matter Physics, Institute of Physics, Chinese Academy of Sciences, Beijing, China; [4]Department of Neuroscience, Yale University School of Medicine, New Haven, United States; [5]Howard Hughes Medical Institute, Yale University School of Medicine, New Haven, United States; [6]Program in Cellular Neuroscience, Neurodegeneration and Repair, Yale University School of Medicine, New Haven, United States; [7]College of Optical and Electronic Technology, China Jiliang University, Hangzhou, China; [8]Integrated Graduate Program in Physical and Engineering Biology, Yale University, New Haven, United States; [9]Department of Cellular and Molecular Physiology, Yale University School of Medicine, New Haven, United States; [10]Nanobiology Institute, Yale University, West Haven, United States; [11]Department of Chemical Engineering, Columbia University, New York, United States; [12]Kavli Institute for Neuroscience, Yale University School of Medicine, New Haven, United States; [13]Department of Molecular Biophysics and Biochemistry, Yale University, New Haven, United States; [14]Laboratoire de Neurophotonique, Faculté des Sciences Fondamentales et Biomédicales, Centre National de la Recherche Scientifique (CNRS) UMR 8250, Université Paris Descartes, Paris, France

*For correspondence:
pietro.decamilli@yale.edu (PDC);
erdem.karatekin@yale.edu (EK);
yongli.zhang@yale.edu (YZ)

†These authors contributed equally to this work

Competing interests: The authors declare that no competing interests exist.

**Abstract** Many biological processes rely on protein–membrane interactions in the presence of mechanical forces, yet high resolution methods to quantify such interactions are lacking. Here, we describe a single-molecule force spectroscopy approach to quantify membrane binding of C2 domains in Synaptotagmin-1 (Syt1) and Extended Synaptotagmin-2 (E-Syt2). Syts and E-Syts bind the plasma membrane via multiple C2 domains, bridging the plasma membrane with synaptic vesicles or endoplasmic reticulum to regulate membrane fusion or lipid exchange, respectively. In our approach, single proteins attached to membranes supported on silica beads are pulled by optical tweezers, allowing membrane binding and unbinding transitions to be measured with unprecedented spatiotemporal resolution. C2 domains from either protein resisted unbinding forces of 2–7 pN and had binding energies of 4–14 $k_B$T per C2 domain. Regulation by bilayer composition or $Ca^{2+}$ recapitulated known properties of both proteins. The method can be widely applied to study protein–membrane interactions.

DOI: https://doi.org/10.7554/eLife.30493.001

## Introduction

Protein–membrane interactions play pivotal roles in numerous biological processes, including membrane protein folding (*Yu et al., 2017*; *Popot and Engelman, 2016*; *Min et al., 2015*), lipid metabolism and transport (*Giordano et al., 2013*; *Reinisch and De Camilli, 2016*; *Hammond and Balla,*

*2015*; *Wong et al., 2017*), membrane trafficking (*Zhou et al., 2017*; *Pérez-Lara et al., 2016*; *Wu et al., 2017*; *Hurley, 2006*; *Shen et al., 2012*; *McMahon and Gallop, 2005*), signal transduction (*Dong et al., 2017*; *Aggarwal and Ha, 2016*; *Lemmon, 2008*; *Das et al., 2015*), and cell motility (*Wang and Ha, 2013*; *Tsujita and Itoh, 2015*). Studying these interactions is often difficult, especially when they involve multiple intermediates, multiple ligands, mechanical force, large energy changes, or protein aggregation (*Dong et al., 2017*; *Pérez-Lara et al., 2016*; *Arauz et al., 2016*). Traditional experimental approaches based on an ensemble of protein molecules often fail to reveal the intermediates, energetics, and kinetics of protein–membrane binding, due to difficulties in synchronizing the reactions and in applying force to proteins or membranes (*Zhang et al., 2013*). Single-molecule methods can overcome these problems, and have been applied to study dynamics and folding of numerous soluble proteins and an increasing number of membrane proteins (*Knight et al., 2010*; *Knight and Falke, 2009*; *Vasquez et al., 2014*; *Aggarwal and Ha, 2016*; *Erkens et al., 2013*; *Munro et al., 2014*; *Min et al., 2015*; *Yu et al., 2017*). However, high resolution single-molecule methods to probe protein–membrane interactions in the presence of force are lacking.

Here, we used optical tweezers (OTs) to measure both the force, affinity, and kinetics associated with interactions between single proteins and lipid bilayers. OTs use tightly focused laser beams to trap silica or polystyrene beads in a harmonic potential (*Zhang et al., 2013*) (*Figure 1A*). The beads act as force and displacement sensors while applying tiny forces (0.02–250 pN) to single molecules attached to the beads. An optical interference method detects the bead positions (*Gittes and Schmidt, 1998*). High-resolution OTs achieve extremely high spatiotemporal resolution (~0.3 nm, ~20 μs) in a range of force that can reversibly unfold a biomolecule or is generated by molecular motors (*Abbondanzieri et al., 2005*; *Neupane et al., 2016*; *Moffitt et al., 2006*; *Zhang et al., 2013*; *Gao et al., 2012*; *Cecconi et al., 2005*; *Bustamante et al., 2004*). Recently, OTs have been used to test membrane binding of the vesicle tethering complex EEA1 (*Murray et al., 2016*). However, to our knowledge, OTs have not been applied to measure both the energy and the detailed kinetics of protein–membrane interactions. A major objective of our work is to establish a general approach based on OTs to quantify membrane-binding energy, kinetics, and accompanying force production, using the C2 domains of synaptotagmin 1 (Syt1) (*Brose et al., 1992*; *Südhof, 2013*; *Chapman, 2008*) and extended synaptotagmin 2 (E-Syt2) (*Min et al., 2007*; *Giordano et al., 2013*; *Schauder et al., 2014*) as model proteins.

Synaptotagmins (Syts) and extended synaptotagmins (E-Syts) share similar modular structures (*Min et al., 2007*; *Reinisch and De Camilli, 2016*; *Gustavsson and Han, 2009*), including an N-terminal membrane anchor and two to five C-terminal C2 domains, with an additional synaptotagmin-like mitochondrial membrane protein (SMP) module in the case of the E-Syts (*Alva and Lupas, 2016*; *Schauder et al., 2014*) (*Figure 1B*). The C2 domain is one of the most abundant and highly conserved membrane-binding modules, with ~200 C2 domains encoded by the human genome (*Lemmon, 2008*; *Corbalan-Garcia and Gómez-Fernández, 2014*). Their binding to membranes is regulated by the phospholipid composition of the bilayer and in many cases is $Ca^{2+}$-dependent (*Südhof, 2013*; *Monteiro et al., 2014*). The Syt family comprises at least 15 proteins and contain two cytosolic C2 domains. They are anchored to secretory organelles, including neuronal synaptic vesicles, and help mediate their interactions with the plasma membrane (*Südhof, 2013*; *Gustavsson and Han, 2009*; *Chapman, 2008*; *Pérez-Lara et al., 2016*). They act as $Ca^{2+}$ sensors that cooperate with soluble N-ethylmaleimide-sensitive factor attachment receptors (SNAREs) to mediate $Ca^{2+}$-triggered exocytosis, leading to release of neurotransmitters, peptide hormones and a variety of other molecules (*Südhof and Rothman, 2009*; *Zhou et al., 2017*; *Chapman, 2008*). Different from Syts, E-Syts are located on the endoplasmic reticulum membrane and contain five (for E-Syt1) or three (for E-Syt2 and E-Syt3) C2 domains in addition to the SMP domain (*Min et al., 2007*; *Giordano et al., 2013*). E-Syt C2 domains regulate lipid transfer, instead of membrane fusion, between the endoplasmic reticulum and the plasma membrane via the SMP domain (*Giordano et al., 2013*; *Saheki et al., 2016*; *Yu et al., 2016*; *Schauder et al., 2014*). Therefore, both Syts and E-Syts bind membranes in trans through their multiple C2 domains and are well-positioned to generate force to draw two membranes into proximity required for membrane fusion or lipid exchange (*van den Bogaart et al., 2011*; *Lin et al., 2014*; *Krishnakumar et al., 2013*). However, forces generated by Syts and E-Syts and membrane-binding dynamics under load have not been quantified.

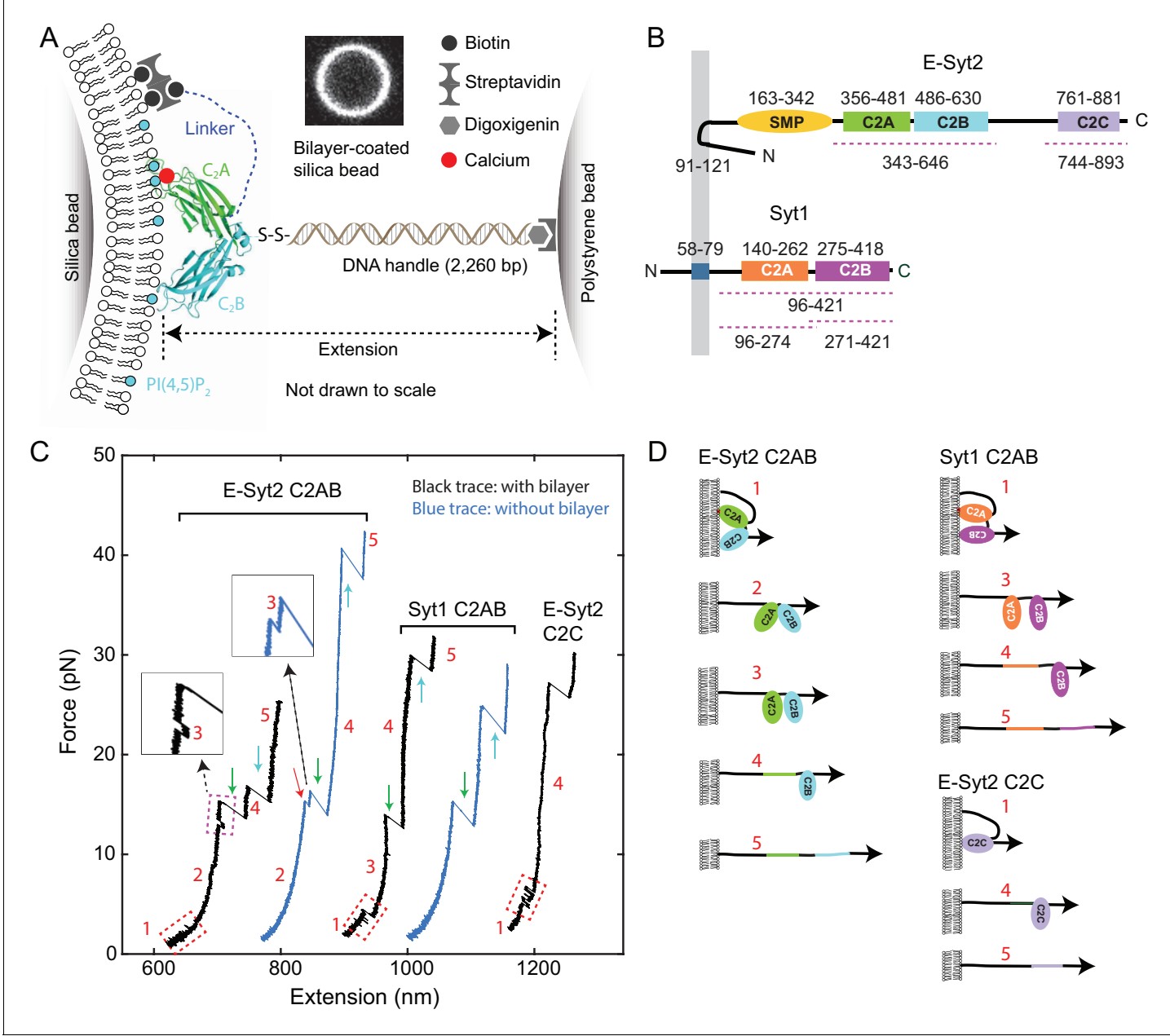

**Figure 1.** Experimental setup to study membrane binding and unfolding of C2 domains and interactions between C2 domains. (**A**) Schematic diagram of the experimental setup to pull a single C2 domain (E-Syt2 C2AB shown) anchored on a lipid bilayer supported on a silica bead. The inset shows the fluorescence image of the bilayer on a silica bead of 5 µm in diameter. (**B**) Domain diagrams of Syt1 and E-Syt2. The dashed lines mark the truncated domains used in this study. (**C**) Force-extension curves (FECs) obtained by pulling C2 domains in the presence of supported bilayers (black) or in its absence (blue). Red-dashed rectangles mark reversible membrane binding and unbinding, while the cyan dashed rectangle indicates reversible C2AB association and dissociation. Green and cyan arrows mark unfolding of C2A and C2B domains, respectively. The insets show the transient state 3. Throughout the text, the FECs were mean-filtered to 100 Hz and shown. The E-Syt2 C2AB was pulled in the presence of membranes composed of 75% POPC, 20% DOPS, 5% PI(4,5)P$_2$, and 0.03% biotin-PEG-DSPE. E-Syt2 C2C and Syt1 C2AB were tested on membranes with a similar composition, except for a decrease in DOPS to 10% and a corresponding increase in POPC to 85%. The solution contained 25 mM HEPES (pH 7.4), 200 mM NaCl, and 100 µM Ca$^{2+}$ for E-Syt2 C2AB and Syt1 C2AB or no Ca$^{2+}$ for E-Syt2 C2C. (**D**) Diagram of different C2 domain states derived from the FECs: 1, membrane-bound state; 2, unbound state with two associated C2 domains; 3, unbound state with two dissociated C2 domains; 4, state with a single folded C2 domain; 5, fully unfolded state.

DOI: https://doi.org/10.7554/eLife.30493.002

The following figure supplements are available for figure 1:

**Figure supplement 1.** Procedures to make membrane-coated silica beads.

*Figure 1 continued on next page*

*Figure 1 continued*

DOI: https://doi.org/10.7554/eLife.30493.003

**Figure supplement 2.** Lipid bilayers supported on silica beads are uniform and mobile.

DOI: https://doi.org/10.7554/eLife.30493.004

**Figure supplement 3.** Histogram distributions of the unfolding forces (top panel) and extension increases (bottom panel) associated with E-Syt2 C2A and C2B unfolding.

DOI: https://doi.org/10.7554/eLife.30493.005

**Figure supplement 4.** Histogram distributions of the unfolding forces (top panel) and extension increases (bottom panel) associated with Syt1 C2A and C2B unfolding.

DOI: https://doi.org/10.7554/eLife.30493.006

**Figure supplement 5.** Force-extension curves (FECs) obtained by pulling Syt1 C2AB, C2A or C2B domain anchored on the supported bilayers (black) or on the streptavidin-coated beads without membranes (blue) in the presence ('+') or absence of ('-') of 100 μM Ca$^{2+}$ in the solution.

DOI: https://doi.org/10.7554/eLife.30493.007

**Figure supplement 6.** Extension-time trajectories (black) of Syt1 C2AB domain pair or individual C2A and C2B domains at the indicated constant mean forces and calcium concentrations and their idealized transition (red) derived from hidden-Markov modeling.

DOI: https://doi.org/10.7554/eLife.30493.008

**Figure supplement 7.** Force-dependent unbinding probabilities (top) and transition rates (bottom) of Syt1 C2B domain.

DOI: https://doi.org/10.7554/eLife.30493.009

We have developed a single-molecule assay based on high-resolution OTs to measure the force, energy, and kinetics of membrane binding by C2 domains of Syt1 and E-Syt2. Our method can be generally applied to study complex protein–membrane interactions with unprecedented spatiotemporal resolution.

## Results

### Experimental setup

High-resolution dual-trap OTs pull a single molecule tethered between two beads, forming a dumbbell in solution suspended by optical traps (*Figure 1A*). To introduce membranes to the dumbbell system, we coated a silica bead with a lipid bilayer (*Bayerl and Bloom, 1990*; *Brouwer et al., 2015*; *Murray et al., 2016*), using a protocol outlined in *Figure 1—figure supplement 1*. Lipids in the supported bilayer are mobile, uniformly distributed around the bead surface, and free of visible defects, as reported previously (*Baksh et al., 2004*; *Brouwer et al., 2015*) (*Figure 1A*, inset, and *Figure 1—figure supplement 2*). Previous experiments showed that an excessive amount of membrane could be added to bead surfaces under different coating conditions, especially in high salt concentration (*Murray et al., 2016*; *Pucadyil and Schmid, 2008*). The floppy membrane would detach from silica surfaces upon pulling, complicating data analysis. To increase the mechanical stability of membranes, we coated membranes on silica beads in a solution at physiological ionic strength and 37°C, and performed the pulling experiments at room temperature (~23°C). The temperature decrease reduces the area per lipid in the bilayer (*Petrache et al., 2000*), which removes the possible excess membrane on the bead surface.

To probe C2 domain–membrane interactions, we used C2 domains from E-Syt2 and Syt1 as model domains, as previous studies are available for comparison. For E-Syt2, we separately purified and tested the C2AB domain pair and the C2C domain (*Figure 1B*, regions marked by dashed lines). The C2A and C2B domains of E-Syt2 strongly associate with each other to form a V-shaped structure (*Figure 1A*) and have not been purified separately (*Schauder et al., 2014*; *Xu et al., 2014*). For Syt1 we purified and tested both the C2AB domain pair and the individual C2A and C2B domains (*Figure 1B*). We attached each protein fragment to the lipids in the supported bilayer via a flexible N-terminal peptide linker of 40–81 amino acids (a.a.) through biotin–streptavidin interactions (*Figure 1A*). This stable anchor kept the C2 domain (or C2 domain pair) near the membrane, thus facilitating its rebinding after unbinding and our measurement of protein-binding energy and dynamics. A similar polypeptide linker was used to join two proteins in order to study their interactions by single-molecule force spectroscopy (*Kim et al., 2010*). The C-terminus of the protein fragment was attached via a DNA handle (*Cecconi et al., 2005*; *Jiao et al., 2017*) to a polystyrene bead that was not membrane-coated. To tether a single protein between two beads, we first bound the

C2-DNA conjugate to the polystyrene bead, trapped this bead and brought it close to the trapped bilayer-coated silica bead to allow binding of the protein to the supported bilayer via both the biotinylated N-terminus (stable anchor) and the C2 domain(s). Subsequently, the C2 domain (or C2 domain pair) was pulled away from the bilayer by separating the two traps at a speed of 10 nm/s (*Figure 1C*) or keeping the trap separation constant (*Figure 2*). We detected the tension and extension of the protein-DNA tether (*Figure 1A*) to derive the energetics and kinetics of C2 binding and conformational changes.

## C2 domain-membrane binding and conformational transitions

We pulled C2 domains in the presence of membranes with various lipid compositions as indicated in the figures or figure legends. Bead pulling yielded force-extension curves (FECs) containing continuous regions and discrete extension flickering or jumps (*Figure 1C*). The former regions were caused by stretching of the DNA handle and of unfolded polypeptides (see the sequences of our protein

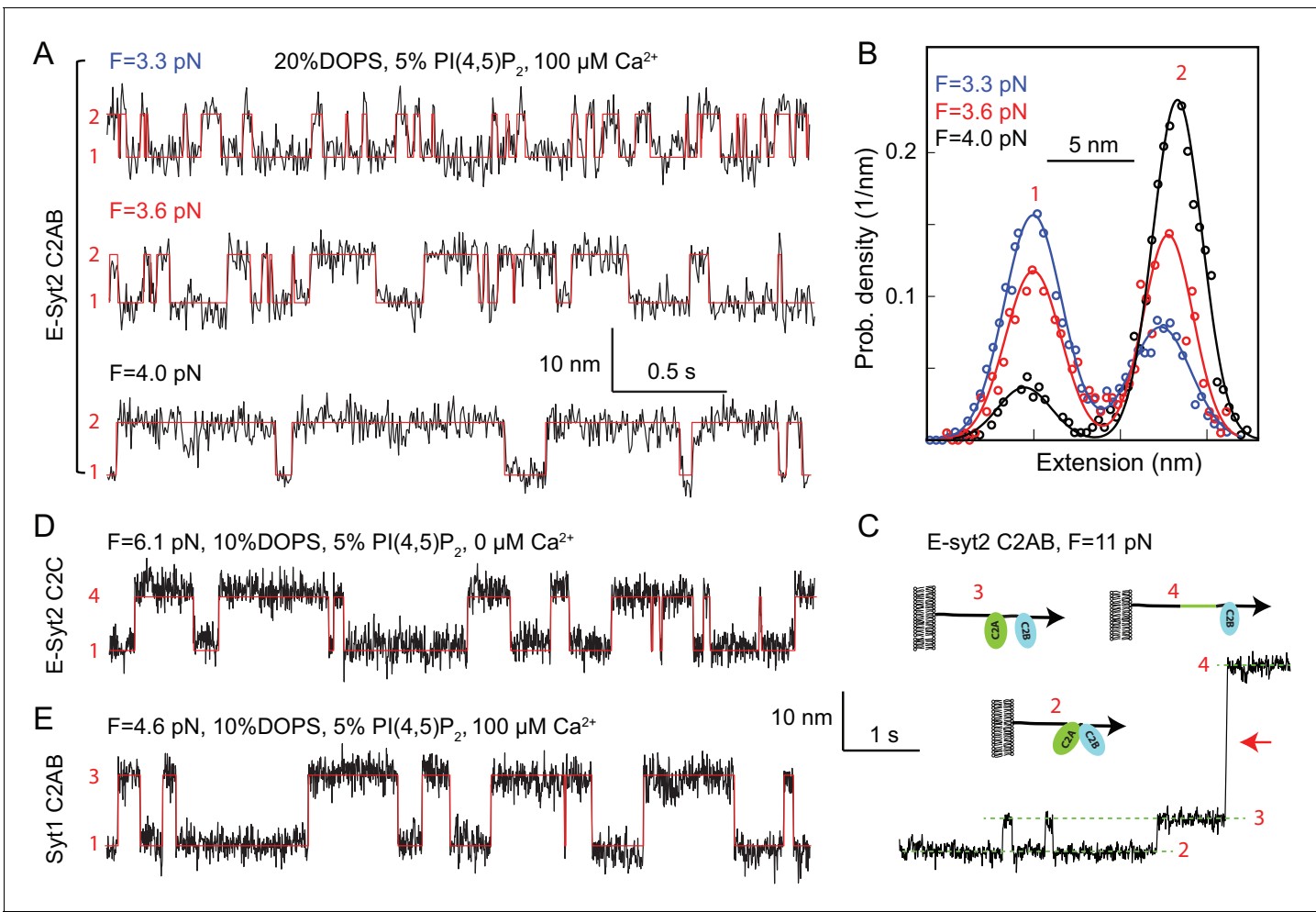

**Figure 2.** Force-dependent reversible membrane binding and unbinding of C2 domains. (**A**) Extension-time trajectories (black) of E-Syt2 C2AB at three indicated constant mean forces (F) and their idealized transitions (red) derived from hidden-Markov modeling (HMM). Positions of different states are marked by their corresponding state numbers as in *Figure 1D*. Throughout the text, the extension-time traces were mean-filtered to 200 Hz and shown. (**B**) Probability density distributions of the extensions (symbols) shown in A and their best-fits by a sum of two Gaussian functions (curves). (**C**) Extension-time trajectory at 11 pN force revealing reversible E-Syt2 C2AB domain dissociation and association before C2A unfolding (red arrow). Different states are marked by green dashed lines and depicted as insets. (**D–E**) Extension-time trajectories (black) of E-Syt2 C2C (**D**) and Syt1 (**E**) at constant forces. Note that the trajectories in C-E share the same extension and time scales.
DOI: https://doi.org/10.7554/eLife.30493.010

constructs in Materials and methods) (*Bustamante et al., 1994*; *Zhang et al., 2013*), while the latter regions were due to C2 domain binding/unbinding or to C2 domain conformational transitions.

With both E-Syt2 C2AB and Syt1 C2AB in the presence of 100 μM $Ca^{2+}$, at 3–5 pN we observed fast extension flickering (*Figure 1C*, regions in the first and third FECs marked by red dashed rectangles), which was better resolved at constant trap separation or mean force (*Figure 2*). The flickering required the presence of the membrane, as it disappeared in the absence of the supported bilayer (*Figure 1C*, blue FECs). Thus, the flickering was caused by reversible C2 domain unbinding from and rebinding to the membrane (*Figure 1D*, transitions between states 1 and 2 for E-Syt2 C2AB and between states 1 and 3 for Syt1 C2AB), a conclusion supported by further experiments described below. As the unbound C2AB domains remained tethered to the membrane via the N-terminal linker sequence, they could rebind the bilayer for forces in this range.

At higher forces 8–13 pN, a small and often reversible jump occurred with E-Syt2 C2AB (*Figure 1C*, region in the first FEC marked by magenta dashed rectangle, and inset). This jump likely represents dissociation of the two C2 domains in the C2AB fragment of E-Syt2 (*Figure 1D*, transition between states 2 and 3 for E-Syt2 C2AB), as the C2A and C2B domains are bound to each other by a stable interface (*Xu et al., 2014*; *Schauder et al., 2014*).

As the force was further increased, for both E-Syt2 C2AB and Syt1 C2AB two larger rips appeared in distinct force ranges, one at 12–22 pN (*Figure 1C*, green arrows), the other at 18–45 pN (cyan arrows) (*Figure 1—figure supplements 3* and *4*). Similar rips were observed in the low and high force ranges when we pulled individual C2A and C2B domains of Syt1, respectively (*Figure 1—figure supplement 5*). These findings suggest that the low and high force rips represent irreversible unfolding of the C2A domains and the C2B domains, respectively, in both E-Syt2 and Syt1 (*Fuson et al., 2009*). They also suggest a lack of a strong association between the C2A and the C2B domains in Syt1, which is in agreement with some reports (*Zhou et al., 2017*; *Vasquez et al., 2014*), although direct interactions between the two domains have been supported by some other studies (*Fuson et al., 2007*; *Liu et al., 2014*). Like the C2AB domains of E-Syt2 and Syt1, E-Syt2 C2C exhibited reversible membrane binding at low force and irreversible membrane-independent unfolding at high force (*Figure 1C*, last FEC). In this case, however, membrane binding was $Ca^{2+}$-independent, consistent with previous studies (*Fernández-Busnadiego et al., 2015*; *Giordano et al., 2013*; *Idevall-Hagren et al., 2015*). In summary, we identified up to five different C2 states in the protein fragments tested, as depicted in *Figure 1D*.

The continuous FEC regions corresponding to the same states in the presence and absence of supported bilayers for all three C2 domains generally overlapped (*Figure 1C*), indicating that the membranes are firmly attached to the silica surfaces and barely contribute to the measured extensions in the force range of interest (<35 pN). However, the FECs appear thicker and are thus noisier in the presence of membranes than in their absence (*Figure 1C*, compare black and blue FECs), possibly due to lateral diffusion of the lipids to which the C2 domains were attached (*Figure 1—figure supplement 2*). Such diffusion was absent when the proteins were directly attached to streptavidin-coated silica beads.

## Energetics and kinetics of C2 domain-membrane binding

To better resolve binding of C2 domains to membranes, we held single proteins at various constant trap separations corresponding to different mean forces (*Jiao et al., 2017*; *Rebane et al., 2016*). In the presence of 100 μM $Ca^{2+}$ and 3–4 pN force, E-Syt2 C2AB reversibly bound to and unbound from membranes, as seen in the extension-time trajectories (*Figure 2A*). The transitions are two-state, as revealed by the two distinct peaks in the corresponding probability density distributions of extension (*Figure 2B*). The width of each peak is determined mainly by Brownian motion of the beads in optical traps (*Jiao et al., 2017*; *Rebane et al., 2016*). Force tilted the equilibrium towards the unbound state, as expected. Inspection of the FEC of the E-Syt2 C2AB domain after membrane unbinding and before C2A domain unfolding also revealed reversible jumps with an average extension change of 5.7 nm at ~11 pN (*Figure 2C*, see also *Figure 1C*). Most likely these jumps reflect dissociation and re-association of C2A and C2B domains of E-Syt2, as predicted by the crystal structure of the C2AB domain pair (*Xu et al., 2014*; *Schauder et al., 2014*).

E-Syt2 C2C also bound to membranes in a two-state manner (*Figure 2D*). However, in contrast to the C2AB domain pairs of E-Syt2, the C2C domain did not need $Ca^{2+}$ for membrane binding and could resist higher pulling forces. This observation is consistent with studies in living cells showing

that the constitutive binding of E-Syt2 to the plasma membrane at resting $Ca^{2+}$ concentrations is mediated by a robust association of its C2C domain with the $PI(4,5)P_2$-rich plasma membrane (*Giordano et al., 2013*; *Idevall-Hagren et al., 2015*).

Syt1 C2AB reversibly bound to membranes in the presence of 100 µM $Ca^{2+}$ similar to E-Syt2 C2AB, but at higher equilibrium force and lower equilibrium rate (*Figure 2E and 3*). Interestingly, although Syt1 C2A and C2B domains barely associate (*Zhou et al., 2017*) as they do in E-Syt2, the two Syt1 C2 domains bound to and unbound from membranes simultaneously within our instrumental resolution, as is indicated by the two-state transition (*Figure 2E*). To dissect contributions of the two Syt1 C2 domains to membrane binding, we tested membrane binding of individual Syt1 C2A and C2B domains, again in the presence of 100 µM $Ca^{2+}$. Whereas C2A domain-membrane interaction was barely discernible under our experimental conditions (*Figure 1—figure supplements 5* and *6*), C2B bound to membranes at a reduced force or affinity compared with the C2AB domain (*Figure 1—figure supplements 6* and *7*). Hence, the two Syt1 C2 domains bind to membranes cooperatively, but the C2B dominates membrane binding, consistent with previous reports (*Bai et al., 2004*; *Pérez-Lara et al., 2016*; *Voleti et al., 2017*).

To quantify the kinetics of C2 domain-membrane binding, we fit the extension trajectories using two-state hidden-Markov modeling (*Zhang et al., 2016b*) (*Figure 2A,D and E*, red lines). The idealized trajectories matched the measured extension trajectories well, revealing the best-fit unbinding probabilities and binding and unbinding rates at each force (*Figure 3* and *Figure 1—figure supplement 7*). As force increases, unbinding probabilities increase in a sigmoidal manner, while unbinding rates increase and binding rates decrease approximately exponentially in the force range tested. All

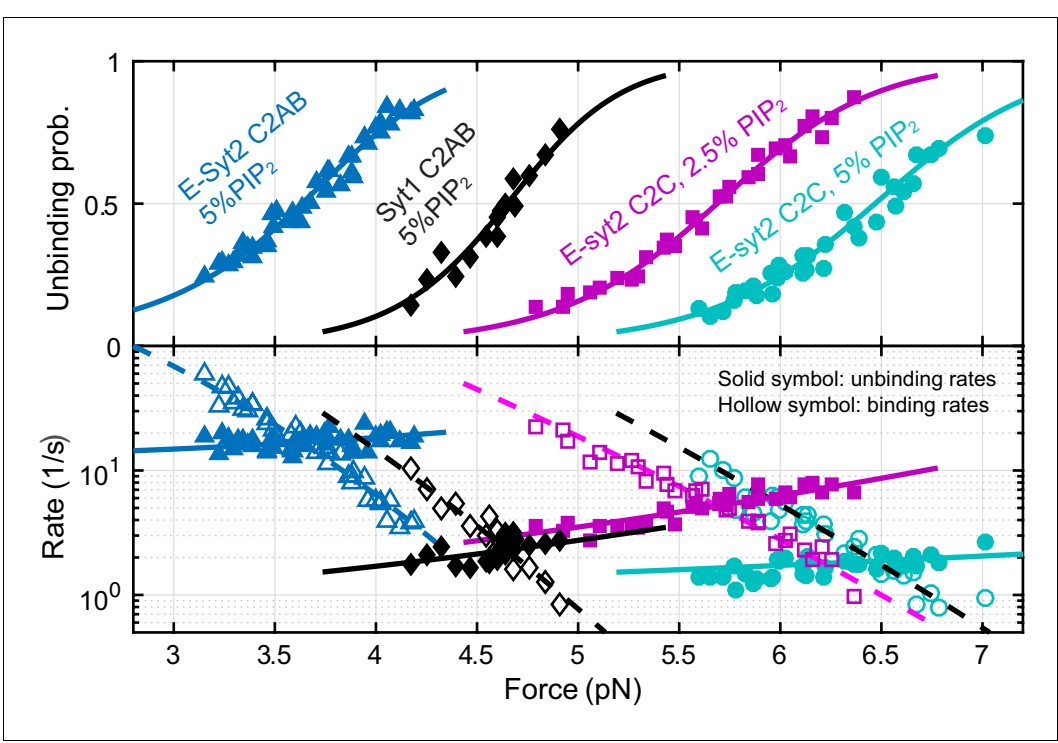

**Figure 3.** Force-dependent unbinding probabilities (top) and transition rates (bottom) and their best model fits (solid and dashed curves) reveal the energy and kinetics of C2 binding at zero force (*Table 1*). Unbinding probabilities and rates are indicated by solid symbols, while binding rates are shown by hollow symbols.

DOI: https://doi.org/10.7554/eLife.30493.011

The following figure supplement is available for figure 3:

**Figure supplement 1.** Diagram illustrating the effect of membrane tethering on protein binding to the membrane.

DOI: https://doi.org/10.7554/eLife.30493.012

these observations are characteristic of two-state transitions (*Bustamante et al., 2004*; *Rebane et al., 2016*).

We simultaneously fit unbinding probabilities, transition rates, and extension changes using a nonlinear model similar to the force-dependent protein folding transitions (*Rebane et al., 2016*), which included effects of the polypeptide linker and the DNA handles on the observed binding and unbinding transitions (see Materials and methods). The model-fitting revealed the best-fit parameters at zero force, including the binding energy, the extension position and energy of the transition state for protein unbinding, and the binding and unbinding rates (*Table 1*). Energy parameters are governed by the first law of thermodynamics: the work to pull the C2 domain away from the membrane is equal to the sum of the unbinding energy of the C2 domain and the entropic energy of the polypeptide link under tension, while the work is determined by the force and extension change associated with the binding and unbinding transition (*Bustamante et al., 2004*; *Zhang et al., 2013*). With membranes composed of 85% POPC, 10% DOPS and 5% PI(4,5)P$_2$, we obtained binding energies of 4.6 $k_B$T for E-Syt2 C2AB, 7.4 $k_B$T for Syt1 C2B, 10.8 $k_B$T for Syt1 C2AB (all in the presence of 100 µM Ca$^{2+}$), and of 12 $k_B$T for E-Syt2 C2C in zero Ca$^{2+}$. The corresponding equilibrium forces, i.e. the forces at which the unbinding probability is 0.5, increase in the same order (2.9 pN, 3.1 pN, 4.7 pN, and 6.5 pN). The increase in the binding energy arises from the corresponding increase in the binding rate ($k_b$) and decrease in the unbinding rate ($k_{ub}$). We estimated the distance between the transition state and the bound state ($\Delta x^{\ddagger}$) to be 1.0 (±0.3, standard deviation) nm for E-Syt2 C2AB, 1.4 (±0.5) nm for Syt1 C2AB, and 0.7 (±0.2) nm for E-Syt2 C2C (see Materials and Methods) (*Bustamante et al., 2004*; *Rebane et al., 2016*). Due to these short distances, the application of force (F) to the proteins only slightly reduces the energy barrier (by ~F × $\Delta x^{\ddagger}$) for protein unbinding (*Rebane et al., 2016*). Consequently, the unbinding rates of these C2 domains at zero force (*Table 1*) are generally within five fold smaller than the corresponding transition rates at the equilibrium force (equilibrium rate), or 43 s$^{-1}$, 3 s$^{-1}$, 3 s$^{-1}$ for E-Syt2 C2AB, E-Syt2 C2C, and Syt1 C2AB, respectively. In contrast, the application of force reduces the binding rates of C2 domains at zero force by >20,000 fold for E-Syt2 C2C and Syt1 C2AB and by ~70 fold for E-Syt2 C2AB under our experimental conditions.

## Effect of membrane tethering on protein-binding energy and kinetics

The binding energy and rates at zero force derived above depend on the membrane tether that keeps the C2 domain near the membrane after unbinding, whereas the unbinding rate is expected to be minimally affected by membrane tethering (*Zhang et al., 2016a*). To examine the effect of membrane tethering, we developed a theory to estimate the binding energy and rates in the absence of the tether.

**Table 1.** Binding energies, binding rates, and unbinding rates of C2 domains at zero force.
The bimolecular binding energies ($E_{on}$) and rates ($k_{on}$) in the absence of membrane tethers were derived from the corresponding energies ($E_b$) and rates ($k_b$) measured by our assay in the presence of membrane tethers by *Equations (9) and (7)*, respectively, whereas the unbinding rates ($k_{ub}$) are independent of membrane tethers. Also shown are the equilibrium forces under which the C2 domains bind to membranes with a probability of 0.5.

| | [Ca$^{2+}$] (µM) | PI(4,5)P$_2$% | DOPS% | Equilibrium force (pN) | Binding energy with tether ($E_b$) ($k_B$T) | Binding energy without tether ($E_{on}$) ($k_B$T) | Log$_{10}$[$k_b$ (s$^{-1}$)] | Log$_{10}$[$k_{on}$ (M$^{-1}$s$^{-1}$)] | Log$_{10}$[$k_{ub}$ (s$^{-1}$)] |
|---|---|---|---|---|---|---|---|---|---|
| E-Syt2 C2AB | 100 | 5 | 10 | 2.9 (0.1) | 4.6 (0.1) | 6.6 (0.1) | 3.5 (0.1) | 4.3 (0.1) | 1.5 (0.1) |
| | 100 | 5 | 20 | 3.6 (0.4) | 7 (1) | 9 (1) | 4.1 (0.3) | 4.9 (0.3) | 1.1 (0.2) |
| E-Syt2 C2C | 0 | 5 | 10 | 6.5 (0.6) | 12 (1) | 14 (1) | 5.2 (0.9) | 6.1 (0.9) | −0.2 (0.2) |
| | 0 | 2.5 | 10 | 5.7 (0.3) | 10.2 (0.6) | 12.5 (0.6) | 4.3 (0.2) | 5.3 (0.2) | −0.1 (0.1) |
| Syt1 C2AB | 100 | 5 | 10 | 4.7 (0.2) | 10.8 (0.8) | 12.8 (0.8) | 4.6 (0.4) | 5.4 (0.4) | 0 (0.2) |
| | 100 | 2.5 | 10 | 3.5 (0.1) | 7.8 (0.2) | 9.8 (0.2) | 4.1 (0.3) | 4.9 (0.3) | 0.7 (0.3) |
| | 100 | 0 | 30 | 3.8 (0.2) | 8.7 (0.3) | 10.7 (0.3) | 4.2 (0.3) | 5.0 (0.3) | 0.4 (0.3) |
| Syt1 C2B | 100 | 5 | 10 | 3.1 (0.2) | 7.4 (0.5) | 9.4 (0.5) | 4.2 (0.2) | 5.1 (0.2) | 1.0 (0.3) |

DOI: https://doi.org/10.7554/eLife.30493.013

We assumed that the linker polypeptide is anchored on one end to streptavidin at a point with a distance $h_0$ away from the membrane while the other end is free (*Figure 3—figure supplement 1*). We chose a coordinate such that the anchoring point and the outer surface of the supported membrane are located at $r_0 = (0, 0, h_0)$ and $z = 0$, respectively. We treated the polypeptide tether using a Gaussian model for the polymer chain (*Dill and Bromberg, 2010*). Accordingly, the effective concentration $c$ of the free end of the chain at a position $r = (x, y, z)$ in the Cartesian coordinate or $(\rho, \phi, z)$ in the cylindrical coordinate is (*Zhang et al., 2016a*)

$$c = \frac{1}{N_A}\left(\frac{3}{4\pi PL}\right)^{\frac{3}{2}}\exp\left(-\frac{3|r-r_0|^2}{4PL}\right),$$ (1)

where $N_A = 6.02 \times 10^{23}$ per mole is the Avogadro constant, $L$ is the contour length of the polypeptide linker, and $P$ is its persistence length. Suppose that the free end of the linker is attached to a protein at a point located at a distance $h_1$ away from the membrane as the protein binds to the membrane, the effective concentration of the free end at the binding site can be expressed as

$$c(\rho, \phi) = \frac{1}{N_A}\left(\frac{3}{4\pi PL}\right)^{\frac{3}{2}}\exp\left(-\frac{3h^2}{4PL}\right)\exp\left(-\frac{3\rho^2}{4PL}\right)$$ (2)

where $h = h_0 - h_1$. Assuming the protein binds to membranes with an intrinsic bimolecular rate constant $k_{on}$, the rate constant that the tethered protein binds to the membrane surface at $(\rho, \phi)$ could be calculated as

$$k_{bp}(\rho, \phi) = k_{on}c(\rho, \phi).$$ (3)

The total protein-binding rate of the tethered protein $k_b$ is the sum of the binding rate over all available binding sites on the membrane. Assuming each lipid acts as an independent binding site as in most protein–membrane binding assays, we could calculate the total binding rate by integrating *Equation (3)* over the whole membrane surface, that is,

$$k_b = \frac{1}{s}\int_0^{+\infty}\rho d\rho\int_0^{2\pi}d\phi k_{bp}(\rho, \phi)$$ (4)

where $s$ is the area per lipid. Note that $k_{bp}$ has a Gaussian distribution with respect to the variable $\rho$, as is shown in *Equation (2)*, which sets a natural upper bound for the integration over $\rho$ in *Equation (4)*. Substituting *Equations (2) and (3)* into *Equation (4)* and performing the integration, we had

$$k_b = k_{on}c,$$ (5)

where

$$c = \frac{1}{sN_A}\left(\frac{3}{4\pi PL}\right)^{\frac{1}{2}}\exp\left(-\frac{3h^2}{4PL}\right)$$ (6)

is the average effective concentration of the tethered protein on the membrane. Therefore, the protein-binding rate in the absence of the membrane tether can be calculated from our measured binding rate in the presence of the membrane tether as

$$k_{on} = \frac{k_b}{c}.$$ (7)

The rate of the protein dissociating from the membrane ($k_{ub}$) is not affected by the membrane tether. Thus, the protein-binding constant in the absence of the tether ($K_{on} = k_{on}/k_{ub}$) is related to the measured binding constant ($K_b = k_b/k_{ub}$) by the following formula

$$K_{on} = \frac{K_b}{c}.$$ (8)

Similarly, the protein-binding energy in the absence of the tether ($E_{on}$) could be calculated from the binding energy in the presence of the tether ($E_b$) as

$$E_{on} = E_b + \Delta E_c \tag{9}$$

where

$$\Delta E_c = -k_B T \ln\left(\frac{c}{1M}\right). \tag{10}$$

The lengths of the linkers used in our study were 81 a.a. for E-Syt2 C2AB, 40 a.a. for E-Syt2 C2C, 73 a.a. for Syt1 C2AB, and 66 a.a. for Syt1 C2B. To estimate the effective concentrations $c$, we chose the contour length per amino acid as 0.365 nm, the peptide persistence length $P = 0.6$ nm, and the area per lipid $s = 0.7$ nm$^2$ (*Kucerka et al., 2005*). The distances of both linker ends to the membrane were estimated to be $h_0 = 6$ nm and $h_1 = 2$ nm based on the sizes of streptavidin, biotin-PEG-DSPE, and the C2 domains (*Figure 3—figure supplement 1*). Our calculations revealed that the effective concentrations of the four constructs, E-Syt2 C2AB, E-Syt2 C2C, Syt1 C2AB, and Syt1 C2B, were 0.14 M, 0.10 M, 0.14 M, and 0.13 M, respectively. Correspondingly, tethering the proteins to membranes in our assay underestimated the binding energy of the four protein fragments by 2.0 $k_B$T, 2.3 $k_B$T, 2.0 $k_B$T, and 2.0 $k_B$T, respectively. Similarly, we obtained the membrane-binding energies of all C2 domains and their binding and unbinding rates (*Table 1*). The corrected binding energy, binding rate, and unbinding rate of Syt1 C2AB measured by us are consistent with the corresponding values recently reported (12.8 vs 13 $k_B$T, 2.9 × 10$^5$ vs 4 × 10$^5$ M$^{-1}$s$^{-1}$, and 1 vs 1 s$^{-1}$) (*Pérez-Lara et al., 2016*).

In our derivation above, for simplicity we have assumed that the membrane does not significantly disturb the Gaussian distribution of the free end shown in *Equation (1)*. To investigate the effect of the membrane boundary on our derivations, we repeated our calculations using a more accurate, as well as more complex, distribution that takes into account the presence of membranes (*Dill, 1990*). We found that the improved distribution did not significantly change our above calculations. The observation is justified by the fact that the membrane attachment point of the linker polypeptide is far away from the membrane surface (6 nm), compared to the fluctuation of the free end around the attachment point, that is, $\sigma = \sqrt{\frac{2PL}{3}} < 3.4$ nm.

Although our assay did not directly detect membrane binding of Syt1 C2A, we could estimate its binding energy based on the binding energies of both C2AB and C2B domains of Syt1. We modeled Syt1 C2AB as individual C2A and C2B domains linked by a 13 a.a. polypeptide linker with no direct interactions between the three. Then, the membrane-binding energy of the Syt1 C2AB domain ($E_{AB}$) could be expressed as the sum of the binding energies of the C2A domain ($E_A$) and of the C2B domain ($E_B$) and a coupling energy due to domain tethering by the linker, that is,

$$E_{AB} = E_A + E_B + k_B T \ln\left(\frac{c}{1M}\right), \tag{11}$$

where $c$ is the effective concentration of one C2 domain on the membrane while the other C2 domain is already bound to the membrane. The concentration was calculated using *Equation (6)* with $L = 4.7$ nm for the contour length of the linker and $h = 0$, yielding $c = 0.69$ M and a coupling energy of $-0.38$ $k_B$T. Using the derived binding energies of the C2AB and C2B domains in the absence of membrane tethering, we estimated a membrane-binding energy for Syt1 C2A domain to be 3.8 (±0.9) $k_B$T under the condition of 100 µM Ca$^{2+}$, 200 mM NaCl, 85% POPC, 10% DOPS, and 5% PI(4,5)P$_2$. The energy is smaller than the binding energy for Syt1 C2A previously measured under different conditions that favored C2A binding:~6.3 $k_B$T with 100 µM Ca$^{2+}$, 100 mM NCl, 75% DOPC, 25% DOPS (*Davis et al., 1999*) and ~11 $k_B$T with 200 µM Ca$^{2+}$, 100 mM KCl, 47.5% DOPC, 47.5% DOPS, 5% dansyl-PE (*Nalefski et al., 2001*; *Voleti et al., 2017*). The high effective concentration justified the two-state binding and unbinding transition observed for Syt1 C2AB, despite minimum direct interaction between the C2A and C2B domains: once one domain binds to the membrane, the other domain is predicted to bind to the membrane within 0.1 ms, the estimated temporal resolution of our assay, given the high-binding rate constants of the C2A and C2B domains (*Davis et al., 1999*; *Nalefski et al., 2001*).

## Effects of Ca$^{2+}$, salt, DOPS, and PI(4,5)P$_2$ on C2 binding

Previous experiments have shown that membrane binding of C2 domains are differentially sensitive to Ca$^{2+}$ concentration, ionic strength, and lipid composition, which is pivotal for the biological functions of the proteins harboring C2 domains (*Yu et al., 2016*; *Saheki et al., 2016*; *Fernández-Busnadiego et al., 2015*; *Giordano et al., 2013*). To characterize the Ca$^{2+}$-dependence of E-Syt2 C2AB binding to membranes, we first observed its reversible membrane binding in 100 μM Ca$^{2+}$ at constant mean force of 3.8 pN (*Figure 4A*, black region). We then flowed a solution with 1 mM EGTA into the microfluidic channel where the protein was being pulled (*Jiao et al., 2017*). In the absence of free Ca$^{2+}$ in the solution, the C2AB domain stayed in the unbound state with a high extension (green region), indicating that C2AB failed to bind to the membrane. Re-introducing a 100 μM Ca$^{2+}$ solution restored the dynamic binding. The effect of Ca$^{2+}$ on the binding dynamics of C2AB was robust and was observed over many cycles of solution changes.

To further explore the Ca$^{2+}$-dependent binding energy and kinetics, we measured the extension-time trajectories at constant mean forces in a range of Ca$^{2+}$ concentrations (*Figure 4B*), determined their corresponding unbinding probabilities and transition rates (*Figure 4—figure supplements 1* and *2*), and derived the C2 binding energy and kinetics at zero force (*Figure 4C*). As the Ca$^{2+}$ concentration increases, the binding energy of E-Syt2 C2AB quickly increases in 50–100 μM Ca$^{2+}$, then plateaus around 100–200 μM Ca$^{2+}$, and further increases in 200–500 μM Ca$^{2+}$. The logarithm of the binding rate changes in a similar manner, whereas the logarithm of the unbinding rate monotonically decreases. Syt1 C2AB exhibits similar Ca$^{2+}$-dependent multi-phase binding energy change. However, Syt1 C2AB starts to bind the membrane at a lower Ca$^{2+}$ concentration than E-Syt2 C2AB and with a rate that does not significantly change in 20–100 μM Ca$^{2+}$ concentration (*Pérez-Lara et al., 2016*). C2 domains are known to exhibit different binding stoichiometry and affinity for Ca$^{2+}$ that are further altered by anionic lipids (*Bai et al., 2004*; *Monteiro et al., 2014*; *Pérez-Lara et al., 2016*). Whereas E-Syt2 C2B does not bind Ca$^{2+}$, the C2A binds up to four Ca$^{2+}$ with dissociation constants ranging from μM to >10 mM (*Xu et al., 2014*). The multiple-phase binding of C2 domains observed by us is consistent with the multi-valent Ca$^{2+}$ binding by E-Syt2 C2AB and Syt1 C2AB (*Xu et al., 2014*; *Chapman, 2008*). Importantly, their Ca$^{2+}$ sensitivity determined from the Ca$^{2+}$-dependent binding energies are consistent with previous reports (*Pérez-Lara et al., 2016*; *Chapman, 2008*; *Idevall-Hagren et al., 2015*).

C2 domain-membrane binding is modulated by electrostatic interactions (*Corbalan-Garcia and Gómez-Fernández, 2014*; *Lemmon, 2008*) and is expected to be sensitive to the ionic strength of the solution. To examine the effect of ionic strength on C2 domain-membrane binding, we doubled or halved the NaCl concentration in the solution and measured the binding energy and kinetics of E-Syt2 C2AB (*Figure 4D,E* and *Figure 4—figure supplement 3*). As NaCl concentration increased, the C2 domain-membrane affinity monotonically decreased. Thus, NaCl at high concentrations shields the electrostatic attractions between C2 and anionic lipids. The affinity decrease is caused by both a decrease in the binding rate and an increase in the unbinding rate (*Figure 4E*).

Our single-molecule assay also detected effects of lipid composition on C2 domain binding as expected. We found that anionic lipids, both DOPS and PI(4,5)P$_2$, are important for membrane binding by the C2 domains. Reducing the DOPS concentration from 20% to 10% decreased the binding equilibrium force of E-Syt2 C2AB from 3.6 pN to 2.9 pN and the binding energy from 7 k$_B$T to 4.6 k$_B$T (*Table 1* and *Figure 5A*, top trace). Similarly, reducing the PI(4,5)P$_2$ concentration from 5% to 2.5% decreased the binding energy of E-Syt2 C2C from ~12 k$_B$T to ~10 k$_B$T and of Syt1 C2AB from ~10.8 k$_B$T to ~7.8 k$_B$T (*Table 1*, *Figure 5B*, top trace, and *Figure 5—figure supplement 1*). Omitting PI(4,5)P$_2$ minimized membrane binding of E-Syt2 C2AB in the presence of up to 30% DOPS (*Figure 5A*, middle trace) and of Syt1 C2AB in the presence of 10% DOPS (*Figure 5B*, middle trace). These findings are consistent with the critical importance of PI(4,5)P$_2$ for membrane binding of both C2AB domains (*Fernández-Busnadiego et al., 2015*; *Giordano et al., 2013*; *Bai et al., 2004*; *Pérez-Lara et al., 2016*). However, increasing the DOPS concentration to 40% for E-Syt2 C2AB (*Figure 5A*, bottom trace) and 30% for Syt1 C2AB (*Figure 5B*, bottom trace) at least partially rescued their membrane binding in the absence of PI(4,5)P$_2$. In particular, Syt1 C2AB tightly bound to the membrane containing 30% DOPS but no PI(4,5)P$_2$ with a binding energy of 8.7 k$_B$T (*Table 1* and *Figure 5—figure supplement 1*). Thus, both DOPS and PI(4,5)P$_2$ modulate membrane binding of C2AB domains in E-Syt2 and Syt1.

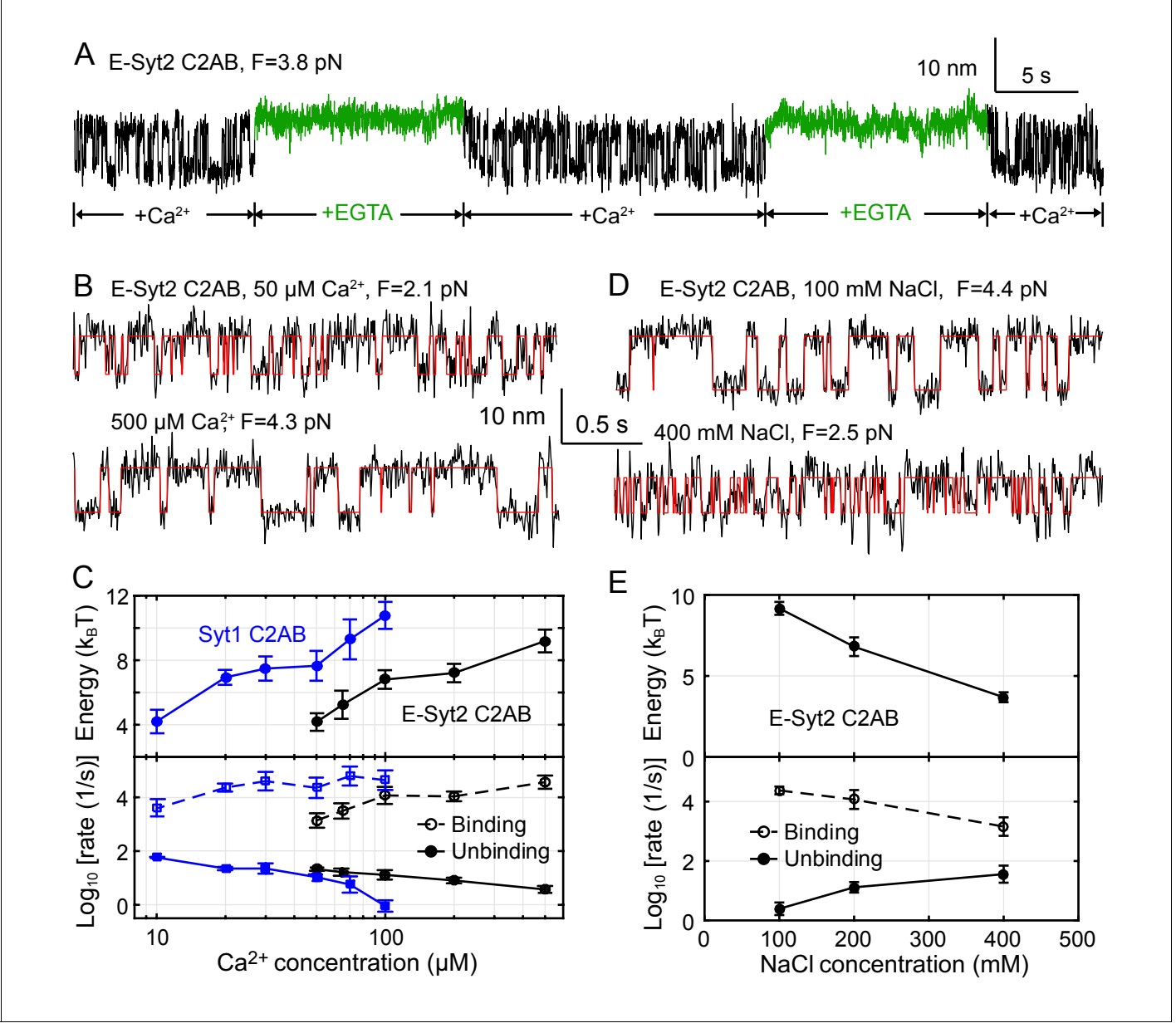

**Figure 4.** Membrane binding of E-Syt2 C2AB and Syt1 C2AB is sensitive to $Ca^{2+}$ and ionic strength. (A) Extension-time trajectories of a single E-Syt2 C2AB domain in the presence of flows of solutions containing either 100 μM $Ca^{2+}$ (black) or 1 mM EGTA (green). (B, D) Extension-time trajectories (black) in different concentrations of $Ca^{2+}$ (B) or NaCl (D) in the solution. Idealized transitions are shown in red lines. Same scales are used in both B and D. (C, E) $Ca^{2+}$-dependent (C) or NaCl-dependent (E) Esyt2-C2AB (black) or Syt1 (blue) binding energy (top) and binding and unbinding rates (bottom).
DOI: https://doi.org/10.7554/eLife.30493.014

The following figure supplements are available for figure 4:

**Figure supplement 1.** Force-dependent unbinding probabilities (top) and transition rates (bottom) of E-Syt2 C2AB measured at 65 μM (blue), 200 μM (black), and 500 μM (purple) $Ca^{2+}$ concentrations.
DOI: https://doi.org/10.7554/eLife.30493.015

**Figure supplement 2.** Force-dependent unbinding probabilities (top) and transition rates (bottom) of Syt1 measured at different $Ca^{2+}$ concentrations.
DOI: https://doi.org/10.7554/eLife.30493.016

**Figure supplement 3.** Force-dependent unbinding probabilities (top) and transition rates (bottom) of E-Syt2 C2AB measured at 100 mM and 400 mM NaCl concentrations.
DOI: https://doi.org/10.7554/eLife.30493.017

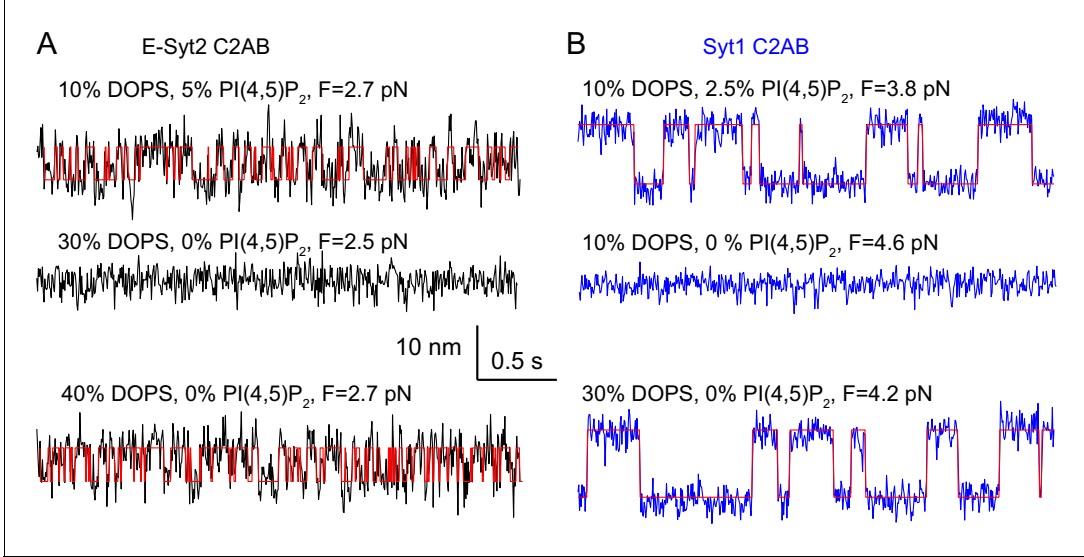

**Figure 5.** Membrane binding of E-Syt2 C2AB and Syt1 C2AB depend on DOPS and PI(4,5)P$_2$. (A–B) Extension-time trajectories of E-Syt2 C2AB (A) or Syt1 C2AB (B) with different concentrations of DOPS or PI(4,5)P$_2$ in the membrane. Idealized transitions are shown in red lines. Same scales are used in both A and B.

DOI: https://doi.org/10.7554/eLife.30493.018

The following figure supplement is available for figure 5:

**Figure supplement 1.** Force-dependent unbinding probabilities (top) and transition rates (bottom) of Syt1 C2AB measured at different lipid compositions as indicated.

DOI: https://doi.org/10.7554/eLife.30493.019

# Discussion

We have developed a new approach to quantify the membrane-binding energy and kinetics of single proteins. In this approach, a protein of interest, in our case a C2 domain or a C2 domain pair, is attached to a lipid bilayer supported by a silica bead and pulled using OTs. Reversible protein–membrane binding is detected based on the associated extension changes with high spatiotemporal resolution, thus allowing us to derive binding affinity and kinetics as a function of force, soluble factors, and lipid compositions.

We chose to apply this new approach to C2 domains of E-Syt2 and Syt1, since previous bulk and (in the case of Syt1) some single-molecule measurements exist for comparison. Overall, our measurements are largely consistent with previous reports, but also yielded additional information not available from bulk or other single-molecule approaches, as explained below.

For E-Syt2, we found that its C2C domain binds to membranes strongly in a PI(4,5)P$_2$-dependent, but calcium-independent manner. This is consistent with observations in intact and semi-intact cells showing that constitutive ER-plasma membrane contacts mediated by E-Syt2 depend upon the Ca$^{2+}$-independent binding of E-Syt2 C2C to PI(4,5)P$_2$ in the plasma membrane (*Idevall-Hagren et al., 2015*; *Giordano et al., 2013*). We have also observed that, in contrast, membrane binding by E-Syt2 C2AB depends on Ca$^{2+}$, again in agreement with other reports (*Min et al., 2007*; *Xu et al., 2014*). As suggested by results of a recent study, Ca$^{2+}$-dependent interaction of E-Syt2 C2AB with the membrane may relieve an autoinhibitory intramolecular interaction between C2AB and the SMP domain (the lipid transport module of E-Syt2) that prevents E-Syt2 from transferring lipids (Xin Bian and Pietro De Camilli, unpublished results).

Recapitulating previous reports (*Min et al., 2007*; *Idevall-Hagren et al., 2015*; *Giordano et al., 2013*), we found that Syt1 C2AB binds to membranes in a Ca$^{2+}$-, DOPS-, and PI(4,5)P$_2$-dependent manner. Our measured binding affinity, binding and unbinding rates in the presence of 100 µM Ca$^{2+}$, 10% DOPS, and 5% PI(4,5)P$_2$ matched well previous measurements under similar experimental conditions. Unlike some reports, however, our assay did not detect any significant interactions between Syt1 C2AB and membranes in the absence of Ca$^{2+}$. The Ca$^{2+}$-independent membrane

binding of Syt1 C2AB was demonstrated by liposome precipitation or sedimentation (see also below about Syt1 C2A domain binding as revealed by TIRFM), but not by other bulk assays conducted by the same groups (*Bai et al., 2004*; *Pérez-Lara et al., 2016*). More work is required to resolve this difference, but it is possible that $Ca^{2+}$-independent membrane binding of Syt1 C2AB may be too weak to be detected in our assay. Alternatively, binding may require cooperation of multiple Syt1 proteins (*Wang et al., 2014*), while our assay assesses the properties of single molecules.

The generally good agreement between our measurements and previous reports validated the new approach. However, the appeal of our single-molecule approach is that it provides additional information not available by bulk measurements. First, indistinguishable bulk kinetics can be produced by different underlying molecular mechanisms, as is well-known for ion channels (*Colquhoun and Hawkes, 2009*) and polymer reactions (*de Gennes, 1982*; *O'Shaughnessy, 1993*). In contrast, single-molecule measurements are advantageous in dissecting complex reaction networks (*Zhang et al., 2013*; *Gao et al., 2012*). Second, we measured binding–unbinding rates under controlled load. Proteins that tether membranes and/or are involved in membrane fusion and fission reactions must work under load, but measuring how membrane binding-unbinding depends on load has been challenging. Third, from the force-dependent binding-unbinding rates, we estimate the position of the transition state for unbinding, only ~1 nm away from the bound states. Fourth, our ability to apply forces with well-defined orientation coupled with extension measurements allow us to detect intramolecular transitions or subunit binding-unbinding. This advantage will be more apparent when proteins with more subunits are probed.

For both E-Syt2 and Syt1, we have found that the forces generated by C2 domain binding to membranes are in the range of 2 to 7 pN, comparable to the forces generated by many motor proteins such as kinesin and myosin (*Zhang et al., 2013*), but are much lower, for example, than the forces produced by neuronal SNARE zippering (~17 pN) (*Ma et al., 2015*; *Gao et al., 2012*). Interestingly, the unbinding rate is less sensitive to force than the binding rate, as is indicated by the smaller slope of the logarithm of the unbinding rate than the slope for the binding rate. This difference suggests that the bound C2 states are tightly confined near membrane surfaces (*Bustamante et al., 2004*; *Rebane et al., 2016*). In other words, the positions of the transition states for unbinding are close to those of the bound states. These observations are consistent with the role of both Syt1 and E-Syt2 in bridging two apposed membranes under tension.

Our study shows that for both E-Syt2 and Syt1, single or double C2 domains bind to membranes with lifetimes shorter than three seconds at zero force and with even shorter lifetimes in the presence of external forces. These lifetimes are much shorter than the lifetimes of docked vesicles and membrane contact sites observed in cells. Nevertheless, membrane contacts observed in cells are mediated by multiple Syt1 and E-Syts, which may dimerize or oligomerize (*Wang et al., 2014*; *Schauder et al., 2014*; *Giordano et al., 2013*), as well as by additional proteins. Thus, long-lived docking of membranes by Syt1 or E-Syts is a consequence of cooperativity among multiple molecules. In support of this idea, an extremely high force of ~425 pN is required to pull apart a single membrane contact site between chloroplast and endoplasmic reticulum (*Andersson et al., 2007*).

Our method complements other single-molecule methods to detect protein–membrane interactions, such as those based on total internal reflection fluorescence microscopy (TIRFM) and atomic force microscopy (AFM). Indeed these two approaches have been used to study interactions between Syt1 C2 domains and membranes. Single fluorophore-labeled proteins can be imaged by TIRFM on supported bilayers, revealing protein binding to, and diffusion in, the bilayers (*Knight et al., 2010*; *Knight and Falke, 2009*; *Vasquez et al., 2014*). The advantages of this method are sensitivity to weaker protein–membrane binding, such as membrane binding of individual Syt1 C2A domains (*Campagnola et al., 2015*). However, force-dependent protein conformational transitions and association–dissociation of different subunits are not measured and the temporal dynamic range is much smaller: 0.05–20 s afforded by the TIRFM method compared to $10^{-4}$-$10^3$ s attained by our method (*Knight and Falke, 2009*; *Zhang et al., 2013*). Because of these differences, our method is well-suited to studying complex multi-stage protein–membrane interactions and related protein conformational transitions. AFM has been used to pull single Syt1 C2AB domains from lipid bilayers (*Takahashi et al., 2010*). However, the C2AB domains were irreversibly detached from bilayers by using loading rates more than a thousand-fold larger than the loading rate used here. Thus, while the study by *Takahashi et al. (2010)* was useful in comparing

detachment forces between mutant and wild-type C2AB domains under high load, binding energies and kinetics could not be measured under the far-from equilibrium conditions employed.

In conclusion, the method described here based on optical tweezers expands the repertoire of techniques that can be used to study protein binding at the single molecule level, is highly versatile and can be applied to study in a comprehensive way complex protein–membrane interactions.

## Materials and methods

### Dual-trap high-resolution optical tweezers

The optical tweezers were home-built and assembled on an optical table in an acoustically-isolated and temperature-controlled room as previously described (*Moffitt et al., 2006*; *Sirinakis et al., 2012*). Briefly, a single laser beam of 1064 nm from a solid-state laser (Spectra-Physics, J20I-BL-106C) is attenuated, expanded ~5 fold in diameter, collimated, and split into two beams with orthogonal polarizations by a polarizing beam splitter. One of the beams is reflected by a mirror attached to a piezoelectrical actuator that turns the mirror along two axes with high resolution (Nano-MTA2, Mad City Labs, WI). The two beams are then combined by another polarizing beam splitter, further expanded two fold by a telescope, and focused by a water immersion 60X objective with a numerical aperture of 1.2 (Olympus, PA) to form two optical traps. The position of one trap can be shifted in the sample plane by turning the actuator-controlled mirror. The outgoing laser light of both traps is collected and collimated by a second objective of the same type, separated based on polarization, and projected to two position sensitive detectors (PSDs, Pacific Silicon Sensor, CA), which detect positions of the two beads through back-focal-plane interferometry (*Gittes and Schmidt, 1998*). The voltage signals from the PSDs are recorded and linearly converted to displacements of the trapped beads and the extension and tension of the protein-DNA tether. The conversion constants, including trap stiffness, are determined by Brownian motions of the trapped beads. Data were acquired at 20 kHz, mean-filtered to 10 kHz, and stored on hard-disks for further analysis. We used a microfluidic flow cell containing three parallel channels to deliver beads through the top and bottom channels or to trap beads in the central channel (*Jiao et al., 2017*). The top and bottom channels are connected to the central channel through glass tubing.

### Single-molecule experiments

Sample preparations for single molecule experiments have been detailed elsewhere (*Jiao et al., 2017*). Briefly, the C2 constructs were reduced by tris-2-carboxyethyl phosphine (TCEP), desalted, mixed with 2,2'-dithiodipyridine-treated DNA handle at a molar ratio of 40:1, and crosslinked to DNA at 4°C overnight by air oxidization. An aliquot of the mixture was bound to anti-digoxigenin antibody-coated polystyrene beads 2.1 µm in diameter (Spherotech, IL) and injected to the top microfluidic channel. The membrane-coated silica beads were injected to the bottom channel. The pulling experiments were performed in 25 mM HEPES, pH 7.4, 0–500 µM CaCl$_2$, and 100–400 mM NaCl, supplemented with an oxygen-scavenging system at room temperature (*Jiao et al., 2017*). The beads were trapped and brought close to form tethers between two bead surfaces.

### Sequences and preparation of protein and DNA samples

Rat Syt1 and human E-Syt2 sequences were used, with intrinsic cysteine in both proteins mutated to either alanine or serine, except for a buried cysteine residue in E-Syt2 C2C (PDBID: 2DMG). The amino acid sequences of all the C2 domain constructs are shown below. The number in parenthesis after each construct name indicate the amino acid numbering in the original protein sequence. Different sequence motifs are colored as follows: Avi-tags in red, the extra linkers in blue, the coding sequences of Syt1 or E-Syt2 in black, the C-terminal cysteine for DNA crosslinking in bold, and the extra C-terminal sequences in purple. The mutated amino acids are underlined.

Syt1 C2AB (96–421, C277A):
GTGLNDIFEAQKIEWHELEGGKNAINMKDVKDLGKTMKDQALKDDDAETGLTDGEEKEEPKEEEKLGK
LQYSLDYDFQNNQLLVGIIQAAAELPALDMGGTSDPYVKVFLLPDKKKKFETKVHRKTLNPVFNEQFTFK
VPYSELGGKTLVMAVYDFDRFSKHDIIGEFKVPMNTVDFGHVTEEWRDLQSAEKEEQEKLGDIAFSLRYVP-
TAGKLTVVILEAKNLKKMDVGGLSDPYVKIHLMQNGKRLKKKKTTIKKNTLNPYYNESFSFEVPFEQIQK

VQVVVTVLDYDKIGKNDAIGKVFVGYNSTGAELRHWSDMLANPRRPIAQWHTLQVEEEVDAMLAVKK
**C**AAAG

### Syt1 C2A (96-274)

GT**GLNDIFEAQKIEWHE**LEGGKNAINMKDVKDLGKTMKDQALKDDDAETGLTDGEEKEEPKEEEKLGK
LQYSLDYDFQNNQLLVGIIQAAAELPALDMGGTSDPYVKVFLLPDKKKKFETKVHRKTLNPVFNEQFTFK
VPYSELGGKTLVMAVYDFDRFSKHDIIGEFKVPMNTVDFGHVTEEWRDLQSAEKEEQEKLG
ELLEGGSG**C**AAAG

### Syt1 C2B (96–140, 271–421, C277A)

GT**GLNDIFEAQKIEWHE**LEGGKNAINMKDVKDLGKTMKDQALKDDDAETGLTDGEEKEEPKEEEEK
LGDIAFSLRYVPTAGKLTVVILEAKNLKKMDVGGLSDPYVKIHLMQNGKRLKKKKTTIKKNTLNPYYNESF
SFEVPFEQIQKVQVVVTVLDYDKIGKNDAIGKVFVGYNSTGAELRHWSDMLANPRRPIAQWHTLQVEEE
VDAMLAVKKELLEGGSG**C**AAAG

### E-syt2 C2AB (343–646, C611S)

**GLNDIFEAQKIEWHE**LEGGSDEGSQGDNGSGDGSKGSGNESGQGTGEGSNGSGDGSGELPWSEVQIA
QLRFPVPKGVLRIHFIEAQDLQGKDTYLKGLVKGKSDPYGIIRVGNQIFQSRVIKENLSPKWNEVYEALVYEH
PGQELEIELFDEDPDKDDFLGSLMIDLIEVEKERLLDEWFTLDEVPKGKLHLRLEWLTLMPNASNLDKVLTDI-
KADKDQANDGLSSALLILYLDSARNLPSGKKISSNPNPVVQMSVGHKAQESKIRYKTNEPVWEENFTFFIH
NPKRQDLEVEVRDEQHQSSLGNLKVPLSQLLTSEDMTVSQRFQLSNSGPNSTIKMKIALRVLHLEKRE
RPPDHQHSAQVKR**C**

### E-syt2 C2C (744-893)

**GLNDIFEAQKIEWHE**GSSHHHHHHSGLVPRGSRLRQLENGTTLGQSPLGQIQLTIRHSSQRNKLIVVVHAC
RNLIAFSEDGSDPYVRMYLLPDKRRSGRRKTHVSKKTLNPVFDQSFDFSVSLPEVQRRTLDVAVKNSGGF
LSKDKGLLGKVLVALASEELAKGWTQWYDLTEDGTRPQAMT**C**

The DNA coding sequence of E-Syt2 C2C construct was cloned into a modified pETDuet-1 vector which has an N-terminal Avi tag, His tag, and a thrombin site. The DNA coding sequences of other constructs were cloned into a modified pET-SUMO vector (Invitrogen, CA) in which the Avi tag was inserted just after the SUMO tag. The plasmids were transformed into *E. coli* BL21(DE3) cells. The cells were grown at 37°C to an OD600 of ~0.6–0.8, induced to express the recombinant proteins with 0.5 mM IPTG at 22°C for 18 hr, and harvested. The proteins were purified first by His60 Nickel Resin (Clontech) and then by gel filtration on a Superdex200 column (GE Healthcare). The purified proteins were biotinylated using biotin ligase (BirA) as described and further purified (*Jiao et al., 2017*). Finally, the proteins were cleaved by the SUMO protease to remove the His-SUMO tags and further cleaned up using Ni-NTA columns.

## Membrane coating on silica beads

The supported lipid bilayers contained different mole percentages of DOPE, DOPS, PI(4,5)P$_2$, and DSPE-PEG(2000)-Biotin as indicated in the text, figures or figure legends. The major steps of bead coating are depicted and described in *Figure 1—figure supplement 1*.

## Hidden-Markov modeling (HMM) and derivations of the energy and kinetics at zero force

Methods and algorithms of HMM and energy and structural modeling are detailed elsewhere (*Zhang et al., 2016b*; *Jiao et al., 2017*; *Rebane et al., 2016*). The MATLAB codes used for these calculations can be found in Ref. (*Gao et al., 2012*) and are available upon request. Briefly, extension-time trajectories at constant trap separations were mean-filtered using a time window of 1–3 ms and then analyzed by HMM. This analysis revealed unbinding probabilities, binding rates, unbinding rates, and extension changes associated with the binding and unbinding transitions at different trap separations. The corresponding idealized state transitions were calculated using the Viterbi algorithm. The average forces for the bound and the unbound states at each trap separation were determined based on the idealized states, whose mean gives the mean force shown in all unfolding

probability and rate plots as a function of force (*Rebane et al., 2016*). We determined the binding energy and binding and unbinding rates at zero force by simultaneously fitting the measured unbinding probabilities, transition rates, and extension changes using a nonlinear model (*Rebane et al., 2016*). In this model, we chose free energies of the bound protein state and the unbinding transition state, the distance of latter state to the membrane at zero force as fitting parameters. Then the free energies of the three states (the bound state, the unbound state, and the transition state) in the presence of force were calculated. These energies represent the total energy of the whole dumbbell system in a given protein-binding state, and additionally include entropic energies of the unfolded polypeptide and the DNA handle due to stretching and potential energy of both trapped beads. The contour length of the stretched polypeptide was state-dependent and chosen as a reaction coordinate. In particular, the linker sequence was counted as part of the reaction coordinate because it was stretched, but not in the bound state because it was no longer stretched. Subsequently, we calculated the unbinding probability based on a Boltzmann distribution and the binding and unbinding rates according to Kramers' theory. Finally, we fit the calculated quantities against their experimental measurements by a nonlinear least-squares method to determine the best-fit parameters. Optical tweezers measure the relative force on the same single molecules with high precision (~0.02 pN) and the absolute force on different molecules with modest accuracy, typically ~10% of the measured force value (*Moffitt et al., 2006*). To improve the accuracy to derive the energy and kinetics of protein binding at zero force, we first determined the average equilibrium force from measurements on 10–130 single molecules, which gives the equilibrium force value shown in *Table 1*. Then the curves of force-dependent unbinding probability and transition rates measured on each molecule were slightly shifted along x-axis so that its equilibrium force matched the average equilibrium force. Subsequently, the nonlinear model fitting was performed to determine the binding energy and rates at zero force. Typically, fitting results from three to eight independent data sets were averaged and reported (*Table 1*).

## Acknowledgements

We thank Aleksander Rebane and Tong Shu for help in data analysis, Grace Chen for help in construct preparation, Joerg Nikolaus for technical assistance, and Jeffrey Knight for discussion. This work was supported by the NIH grants (R01GM093341 and R01GM120193 to YZ, R01NS36251 and DA018343 to PDC, R01GM108954 and R01GM114513 to EK, and the training grant T32GM007223). Research reported in this publication was also supported in part by the Kavli Foundation (PDC and EK), the Brain Research Foundation (YZ), and the Raymond and Beverly Sackler Institute for Biological, Physical and Engineering Sciences at Yale (JJ, EK, and YZ).

## Additional information

### Funding

| Funder | Grant reference number | Author |
| --- | --- | --- |
| National Institute of General Medical Sciences | T32GM007223 | Junyi Jiao |
| Raymond and Beverly Sackler Institute for Biological, Physical and Engineering Sciences, Yale University | Seed Grant | Erdem Karatekin Yongli Zhang Junyi Jiao |
| Kavli Foundation | | Pietro De Camilli Erdem Karatekin |
| National Institutes of Health | R01NS36251 | Pietro De Camilli |
| National Institutes of Health | DA018343 | Pietro De Camilli |
| National Institutes of Health | R01GM108954 | Erdem Karatekin |
| National Institutes of Health | R01GM114513 | Erdem Karatekin |
| Kavli Foundation | Kavli Neuroscience Scholar Award | Erdem Karatekin |

| National Institutes of Health | R01GM093341 | Yongli Zhang |
| National Institutes of Health | R01GM120193 | Yongli Zhang |
| Brain Research Foundation | | Yongli Zhang |

The funders had no role in study design, data collection and interpretation, or the decision to submit the work for publication.

### Author contributions
Lu Ma, Conceptualization, Data curation, Formal analysis, Investigation, Writing—review and editing; Yiying Cai, Conceptualization, Resources, Investigation, Writing—review and editing; Yanghui Li, Data curation, Formal analysis, Validation; Junyi Jiao, Data curation, Formal analysis, Validation, Methodology; Zhenyong Wu, Conceptualization, Resources; Ben O'Shaughnessy, Conceptualization, Formal analysis, Investigation, Writing—review and editing; Pietro De Camilli, Erdem Karatekin, Conceptualization, Supervision, Funding acquisition, Investigation, Writing—review and editing; Yongli Zhang, Conceptualization, Data curation, Software, Formal analysis, Supervision, Funding acquisition, Validation, Investigation, Methodology, Writing—original draft, Writing—review and editing

### Author ORCIDs
Erdem Karatekin http://orcid.org/0000-0002-5934-8728
Yongli Zhang http://orcid.org/0000-0001-7079-7973

### Decision letter and Author response
Decision letter https://doi.org/10.7554/eLife.30493.021
Author response https://doi.org/10.7554/eLife.30493.022

## Additional files

### Supplementary files
• Transparent reporting form
DOI: https://doi.org/10.7554/eLife.30493.020

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
