## [Decision Letter]

Thank you for submitting your article "Single-molecule force spectroscopy of protein-membrane interactions" for consideration by *eLife*. Your article has been reviewed by three peer reviewers, and the evaluation has been overseen by a Reviewing Editor and Randy Schekman as the Senior Editor. The following individual involved in review of your submission has agreed to reveal his identity: Joseph Falke.

The reviewers have discussed the reviews with one another and the Reviewing Editor has drafted this decision to help you prepare a revised submission.

Summary:

This is a potentially important manuscript describing innovative, well-executed studies employing a single molecule, optical tweezer method to study the binding of synatotagmin C2 domains to a supported lipid bilayer. The approach reveals, for the first time, the kinetics and thermodynamics of a single C2 domain binding, unbinding and rebinding the target membrane in many different consecutive steps. The approach enables the first direct analysis of the force dependence of protein-membrane interactions (which is unique, and is physiologically important given the proposed function in membrane fusion which is believed to be mechanical process), and of internal protein interactions. The findings likely also provide information about the zero force limit not previously attainable, although this needs more discussion in the text.

The method is well designed and the results will be of high importance to audiences belonging in many different fields, including single-molecule force study, protein-membrane interactions and membrane trafficking regulated by the C2 domain family.

More specifically, the method is demonstrated using synaptotagmins (syt1 and Esyt2). The authors demonstrate expected dependencies on calcium, sodium chloride, phosphatidylserine and PIP2 for the membrane interactions of these proteins. The measurements of the paper are in agreement with expected behaviors measured using other biochemical methods. For example, the manuscript highlights highly consistent comparisons of their measurements for syt1 to published results (Perez-Lara et al., *eLife* 2016) that used stopped flow FRET observations and ITC studies.

Nevertheless, the manuscript needs to be improved by clearly highlighting what new insights have been obtained using the method that could not have been obtained otherwise and by performing careful comparisons to previous findings of well-studied single domain membrane interactions.

Essential revisions:

1) The new insights in biology claimed in the Discussion are 1) C2 domains withstand 2-7 pN of force in membrane binding; 2) E-syt2 C2C binds membranes in PIP2-dependent, calcium-independent manner; 3) syt1 C2AB binds in a calcium, PS, PIP2-dependent manner. As the paper is being presented as a new method, these findings are not effective in demonstrating the significance of this new methodology because almost all these insights were available by earlier approaches. The method does seem like a worthy experimental target, but the way the results for the particular systems in the paper are presented does not emphasize the relevance of binding in the context of force. This is despite the acknowledged fact that synaptotagmins are involved in membrane fusion reactions that require biological application of forces to membranes.

2) What additional information or advantages does the new approach provide in the zero force limit relative to SM TIRFM studies of protein-membrane binding by 2D diffusion track analysis? The manuscript does not cite (Introduction, first paragraph) the original single molecule studies of protein binding to a supported bilayer via TIRFM monitoring the 2-D diffusion of a fluor-tagged PH domain. The SM diffusion track analysis provides equilibrium affinities, binding and unbinding rates determined by the surface density, by the rate of appearance of single particle tracks, and by the exponential lifetime distribution of the track population, respectively (see PMID 19167305 and 21044585). Subsequent studies of other membrane binding proteins, including at least one synatotagmin isoform (PMID 25437758) provide similar information. All such studies are at zero force.

3) How do the parameters measured by this approach compare to those measured by other methods? If there are SM TIRFM, 2D diffusion track studies of the same Syt and E-Syt C2 domains (seems likely, but I didn't find any) the manuscript should compare the binding affinity, on- and off-rate parameters measured by the new approach and by SM TIRFM diffusion analysis. The new data can also be compared to published bulk stopped flow kinetic parameters for the Syt1 C2A,B domains, as the text briefly mentions (subsection “Energetics and kinetics of C2 domain-membrane binding”, last paragraph); however, a more careful discussion is needed. One would expect that compared to the bulk measurement, the new method, which uses a DNA handle and streptavidin anchor to hold the Syt C2 domains near the target bilayer, should yield higher binding affinity, higher on-rates and similar off-rates (or slower off-rates if microscopic, undetected, rebinding events arising from the tethering are an issue). Surprisingly, the bulk and new parameters are all extremely similar. What is the explanation for this surprising similarity?

4) The text claims that there is no interaction between the Syt1 C2A and C2B domains, but a cursory literature search revealed at least one published study (PMID 24657966) providing AFM evidence for such an interaction. The text should discuss the literature reference(s) proposing the interaction and why the interaction is not detected herein. The cooperative binding and unbinding of Syt1 C2A and C2B noted in the text (subsection “Energetics and kinetics of C2 domain-membrane binding”, third paragraph) may arise from such interactions.

5) Extensive studies in the literature (see for example PMID 11258923) show that Syt1 C2A does in fact bind with high affinity to bilayers, while the present study barely detects this binding. Why is it so weak? The authors should double check the primary structure of their C2A and make sure it is WT.

---

## [Author Response]

Essential revisions:1) The new insights in biology claimed in the Discussion are 1) C2 domains withstand 2-7 pN of force in membrane binding; 2) E-syt2 C2C binds membranes in PIP2-dependent, calcium-independent manner; 3) syt1 C2AB binds in a calcium, PS, PIP2-dependent manner. As the paper is being presented as a new method, these findings are not effective in demonstrating the significance of this new methodology because almost all these insights were available by earlier approaches. The method does seem like a worthy experimental target, but the way the results for the particular systems in the paper are presented does not emphasize the relevance of binding in the context of force. This is despite the acknowledged fact that synaptotagmins are involved in membrane fusion reactions that require biological application of forces to membranes.

A main objective of this study is to report a new method to study protein-membrane interactions. A unique advantage of the method is the possibility of measuring the forces involved in proteinlipid bilayer interactions and the associated protein conformational changes. To our knowledge, the forces generated by membrane binding of C2 domains had not been measured before. Knowledge of these forces is important to gain insight into organelle dynamics within cells. In addition, our method can also measure protein binding energy and kinetics and their associated protein conformational transitions at zero force. In this case, previous results from the widely studied C2 domains helped validate our new approach and it was critical for us to show that results obtained with our method are in overall agreement with results obtained by other techniques.

In response to the comments by the reviewers, we have reorganized the Discussion extensively, trying to clarify the advantages of the approach. In addition, we changed the penultimate paragraph in Introduction and made minor changes throughout the text. Although we can think of several ways force-dependent binding-unbinding rates can affect membrane contact site formation and membrane fusion, we have purposefully avoided speculation in the absence of additional evidence.

Another unique advantage of the method is that binding-unbinding of individual subunits in a polypeptide and/or intramolecular interactions between such subunits can be detected with our approach. Based on the validation of our method reported here, we are beginning to investigate proteins with multiple bilayer binding domains and in preliminary experiments we have observed complex multi-stage C2 domain-membrane binding kinetics. However, these results are still too preliminary to be included here.

2) What additional information or advantages does the new approach provide in the zero force limit relative to SM TIRFM studies of protein-membrane binding by 2D diffusion track analysis? The manuscript does not cite (Introduction, first paragraph) the original single molecule studies of protein binding to a supported bilayer via TIRFM monitoring the 2-D diffusion of a fluor-tagged PH domain. The SM diffusion track analysis provides equilibrium affinities, binding and unbinding rates determined by the surface density, by the rate of appearance of single particle tracks, and by the exponential lifetime distribution of the track population, respectively (see PMID 19167305 and 21044585). Subsequent studies of other membrane binding proteins, including at least one synatotagmin isoform (PMID 25437758) provide similar information. All such studies are at zero force.

We thank the reviewer for bringing to our attention these studies and apologize for the omission. We have added these references, and a fourth one, to the revised manuscript. We have also compared our approach to these alternative approaches to study protein-membrane interactions in Discussion. Overall, these are all complementary methods, with some being more sensitive to weaker interactions (TIRFM), some to stronger ones (AFM and optical tweezers). Ours seems to be the only one where force-dependent equilibrium binding-unbinding rates can be measured. In addition, our method can reveal intramolecular domain association and dissociation whereas this has not been shown by the TIRFM approach.

3) How do the parameters measured by this approach compare to those measured by other methods? If there are SM TIRFM, 2D diffusion track studies of the same Syt and E-Syt C2 domains (seems likely, but I didn't find any) the manuscript should compare the binding affinity, on- and off-rate parameters measured by the new approach and by SM TIRFM diffusion analysis. The new data can also be compared to published bulk stopped flow kinetic parameters for the Syt1 C2A,B domains, as the text briefly mentions (subsection “Energetics and kinetics of C2 domain-membrane binding”, last paragraph); however, a more careful discussion is needed. One would expect that compared to the bulk measurement, the new method, which uses a DNA handle and streptavidin anchor to hold the Syt C2 domains near the target bilayer, should yield higher binding affinity, higher on-rates and similar off-rates (or slower off-rates if microscopic, undetected, rebinding events arising from the tethering are an issue). Surprisingly, the bulk and new parameters are all extremely similar. What is the explanation for this surprising similarity?

Following the reviewer’s suggestion, we have expanded our comparison of previous results to ours on Syt1 and E-Syt2. For Syt1 C2AB, the binding energy and kinetics are consistent with the measurements in recent work (Perez-Lara et al., 2016). To our knowledge, the SM TIRFM has been applied to study Syt1 C2A (Campagnola et al., 2015) and Syt7 C2AB (Vasquez et al., 2014). For E-Syt2 C2AB and C2C, our results are consistent with previous results with respect to calcium and lipid dependence, while we believe we measured the membrane binding energy and kinetics for the first time.

The reviewer is correct in that the membrane tether and the DNA handle used in our assay do affect the binding energy and kinetics of C2 domains. These effects are taken into account in our data analysis. Specifically, these polymers are described using worm-like chain models (Marko and Siggia, 1995;

Bustamante et al., 1994). Consequently, the bimolecular binding energy and kinetics in the absence of the membrane tether and the DNA handle are derived and compared to previous results. We consider this theoretical analysis is an important component of our assay. To emphasize this development, we have moved the relevant description from “Materials and methods” to the main text as a separate section. Table 1 summarizes the binding energies we measured (with the tethers) and the calculated energies after correcting the effect of the tether.

4) The text claims that there is no interaction between the Syt1 C2A and C2B domains, but a cursory literature search revealed at least one published study (PMID 24657966) providing AFM evidence for such an interaction. The text should discuss the literature reference(s) proposing the interaction and why the interaction is not detected herein. The cooperative binding and unbinding of Syt1 C2A and C2B noted in the text (subsection “Energetics and kinetics of C2 domain-membrane binding”, third paragraph) may arise from such interactions.

Our assay did not detect any significant interactions between Syt1 C2A and C2B domains, which is consistent with some of the previous reports, but not others, including the one mentioned by the reviewer. We have added a sentence to our Discussion, citing both opinions. The cooperative binding and unbinding of Syt1 C2A and C2B we observed does not necessarily result from a direct interaction between the two domains.

Even in the absence of a direct interaction, the two tandem C2 domains can bind cooperatively due to the fact that they are linked by a short linker. As one domain binds, it brings the other near the membrane, which helps rapid binding of the second domain. When binding of the second domain occurs sufficiently rapidly after the first domain binds, the two tandem domains appear to bind and unbind simultaneously. This is now discussed in the last paragraph of the subsection “Effect of membrane tethering on protein binding energy and kinetics”. Consistent with this interpretation, our preliminary data suggest that replacing the linker with a proline track does not significantly alter the transition. Nevertheless, we cannot rule out a weak interaction between the two domains that may not be detected by our assay.

5) Extensive studies in the literature (see for example PMID 11258923) show that Syt1 C2A does in fact bind with high affinity to bilayers, while the present study barely detects this binding. Why is it so weak? The authors should double check the primary structure of their C2A and make sure it is WT.

We have double checked our sequence for C2A and other domains and found no mistakes. Using the binding affinities measured for Syt1 C2AB and C2B and theoretical modeling, we estimated the binding energy of C2A to be 3.8 kT, compared to 6.3 kT measured under similar but not identical conditions. We noticed that many experiments showing membrane binding of C2A are carried out under conditions that favor stronger C2A binding, for example, higher percentage of PS and lower concentration of NaCl than those used in our experiments. Our assay did not directly detect C2A binding, probably because the lifetime of C2A binding is short (~4 ms (Davis et al., 1999)), which becomes too brief to be detected by our assay in the presence of force. In comparison, the lifetimes of the C2AB and the C2B binding are 1 s and 0.1 s, respectively. These points are explained in the last paragraph of the subsection “Effect of membrane tethering on protein binding energy and kinetics”.